# Beneficial Effect of Bee Venom and Its Major Components on Facial Nerve Injury Induced in Mice

**DOI:** 10.3390/biom13040680

**Published:** 2023-04-17

**Authors:** Hafsa Er-Rouassi, Meryem Bakour, Soumaya Touzani, Miguel Vilas-Boas, Soraia Falcão, Catherine Vidal, Badiaa Lyoussi

**Affiliations:** 1Centre Borelli, Université de Paris Cité, National Centre for Scientific Research UMR 9010, 75006 Paris, France; 2Laboratory of Natural Substances, Pharmacology, Environment, Modeling, Health, and Quality of Life (SNAMOPEQ), Department of Biology, Faculty of Sciences Dhar Mehraz, Sidi Mohamed Ben Abdellah University, Fez 30000, Morocco; 3The Higher Institute of Nursing Professions and Health Techniques, Fez 30000, Morocco; 4Centro de Investigação de Montanha, Instituto Politécnico de Bragança, Campus de Santa Apolónia, 5300-252 Bragança, Portugal

**Keywords:** peripheral nerve injury, bee venom, melittin, PLA2, facial nerve branches recovery, Swiss mice

## Abstract

Peripheral nerve injury (PNI) is a health problem that affects many people worldwide. This study is the first to evaluate the potential effect of bee venom (BV) and its major components in a model of PNI in the mouse. For that, the BV used in this study was analyzed using UHPLC. All animals underwent a distal section-suture of facial nerve branches, and they were randomly divided into five groups. Group 1: injured facial nerve branches without any treatment. Group 2: the facial nerve branches were injured, and the normal saline was injected similarly as in the BV-treated group. Group 3: injured facial nerve branches with local injections of BV solution. Group 4: injured facial nerve branches with local injections of a mixture of PLA2 and melittin. Group 5: injured facial nerve branches with local injection of betamethasone. The treatment was performed three times a week for 4 weeks. The animals were submitted to functional analysis (observation of whisker movement and quantification of nasal deviation). The vibrissae muscle re-innervation was evaluated by retrograde labeling of facial motoneurons in all experimental groups. UHPLC data showed 76.90 ± 0.13%, 11.73 ± 0.13%, and 2.01 ± 0.01%, respectively, for melittin, phospholipase A2, and apamin in the studied BV sample. The obtained results showed that BV treatment was more potent than the mixture of PLA2 and melittin or betamethasone in behavioral recovery. The whisker movement occurred faster in BV-treated mice than in the other groups, with a complete disappearance of nasal deviation two weeks after surgery. Morphologically, a normal fluorogold labeling of the facial motoneurons was restored 4 weeks after surgery in the BV-treated group, but no such restoration was ever observed in other groups. Our findings indicate the potential of the use of BV injections to enhance appropriate functional and neuronal outcomes after PNI.

## 1. Introduction

Peripheral nerve injury (PNI) is a substantial clinical problem, affecting 300,000 patients in Europe each year [1,2,3]. In contrast to the central nervous system, the peripheral nervous system exhibits intrinsic abilities for regeneration (collateral sprouting, Schwann cell activation, axonal growth) [4]. The consequences of PNI depended on both the severity of the axonal lesion (neurotmesis, axonotmesis, neurapraxia) and the site of the lesion relative to the neuron’s soma. Following a facial nerve lesion, normal facial motricity can be lost, and a complete facial palsy can be observed. Several treatment strategies, such as surgical treatment (neurorrhaphy is the most common method, which involves suturing the proximal and distal stumps of the injured nerve) [5], pharmacological treatment (e.g., corticosteroids and erythropoietin), non-pharmacological treatment (e.g., photobiomodulation, electrical and mechanical stimulations), and cell-based therapies have been proposed to enhance the recovery process after PNI [6,7,8,9,10,11,12]. However, despite there being advanced microsurgical techniques and different treatments available, a complete nerve function recovery has rarely been reported in the literature. Furthermore, sequelae such as synkinesis may also occur. Therefore, it is crucial to develop new potential alternative medical treatments to improve functional recovery, especially in cases of severe injury.

Peripheral motor nerve injury has been shown to elicit several pathological changes such as oxidative stress (OS). A dramatic increase in oxidative-stress-related biomarkers such as nitric oxide synthase (NOS) expression in the affected neuronal cell bodies has been demonstrated [13,14,15,16]. OS is a condition associated with the excess production of reactive oxygen species (ROS) arising through various oxidation pathways and a disruption of the oxidant–antioxidant balance due to a decrease in total antioxidant capacity. It plays a negative role in nerve functional recovery after PNI [17,18]. Medical advances have paved the way for new therapeutic strategies. Among the avenues explored, the application of antioxidant agents has proven effective in preclinical trials on nerve regeneration and function.

The present study was the first to report the potential effect of bee venom (BV) on functional recovery and reinnervation after facial nerve section-suture in mice. BV consists of a complex mixture of more than 18 pharmacologically active compounds, which includes a variety of peptides (including melittin, apamin), enzymes (e.g., phospholipase A2; PLA2), and a high quantity of water (>80%) [19]. Melittin is a major peptide (>50%) component of BV and has been associated with multiple effects such as anti-inflammatory, anti-arthritis, and neuroprotective activities [20,21,22]. The second major component of BV is PLA2, comprising approximately 10–12% of the dry weight of the venom. The PLA2 derived from BV (bvPLA2) belongs to secretory PLA2 (sPLA2) [23,24]. Interestingly, several lines of evidence have indicated the therapeutic effect of bvPLA2 in neurodegenerative diseases, including Parkinson’s and Alzheimer’s diseases. Intraperitoneal administration of bvPLA2 was shown to attenuate learning and memory deficits and exerted anti-neuroinflammation effects in 3xTg-AD mice. It has also been shown that BV can protect dopaminergic neurons from degeneration in experimental PD models with a recovery of locomotor activity [25,26]. Conversely, other BV compounds may have other activities, such as apamin, which can improve neuronal excitability and synaptic plasticity by blocking calcium-activated K+ channels in Alzheimer’s disease [27]. Recently, its neuroprotective effects on cortical neurons have been reported with limited information on the underlying mechanism [28].

Accordingly, BV therapy has been developed to treat various diseases, including inflammatory diseases such as rheumatoid arthritis, osteoarthritis, and neurological diseases such as amyotrophic lateral sclerosis (ALS) and Parkinson’s disease (PD) [23,29,30,31].

This study assessed the protective effect of BV in a model of PNI: section-suture (SS) of the buccal and marginal mandibular branches of the facial nerve (NF) in mice. We had three major aims:-To analyze the effect of BV therapy on behavioral recovery (whisker movement and nasal deviation) after distal section-suture of the facial nerve.-To analyze the recovery of facial motoneuron retrograde FG labeling in treated and control mice after distal section-suture of the facial nerve.-To compare the BV and betamethasone/or the BV and mixture of the bvPLA2 and melittin effects on both functional and neuronal recovery.

## 2. Materials and Methods

### 2.1. Bee Venom and Its Major Components

Bee venom from *Apis mellifera* (L.) bee species was collected in March 2021 by a professional beekeeper from Beni-Mellal, Morocco. The sample was lyophilized and stored until analysis at 4 °C in the dark. The bee venom was harvested by placing a glass plate covered with a food-grade polyethylene film, surmounted by metal wires connected to a potential of 15–20 V. The principle consists of placing the device on the flight board of the bees. After electrical stimulation, the bee stings the glass plate. The food film located above the removable glass bottom allows the bee to withdraw its stinger without dying, and the venom is deposited on the glass plate. The plates were stored in a dry and sterile room for two days, and the food-grade polyethylene film was removed. The glass plates were then scraped with a blade to obtain the venom powder, which was handled with care.

Phospholipase A2 (reference: P9279-1MG) and melittin (reference: M2271-1Mg) from honey bee venom (*Apis mellifera* L.) were purchased from Sigma Aldrich (St. Louis, MO, USA).

### 2.2. UHPLC Peptide Analysis of Honeybee Venom

The UHPLC analyses were performed on a Dionex Ultimate 3000 UPLC instrument (Thermo Scientific, Waltham, MA, USA) equipped with a diode-array detector. The chromatographic system consisted of a quaternary pump, an autosampler maintained at 5 °C, a degasser, a photodiode array detector, and an automatic thermostatic column compartment. The chromatographic separation was carried out on an XSelect CSH130 C18, 100 mm × 2.1 mm id, 2.5 µm XP column (Waters, Milford, MA, USA), and its temperature was maintained at 30 OC. The mobile phase was composed of (A) 0.1% (*v*/*v*) formic acid in water and (B) 0.1% (*v*/*v*) formic acid in acetonitrile, which was previously degassed and filtrated. The LC conditions used were according to previous work [32]. Spectral data for all peaks were accumulated in the range of 190–500 nm. Control and data acquisition was carried out with the Xcalibur^®^ data system (Thermo Scientific). Cytochrome C, as the internal standard (IS), was prepared in deionized water at a concentration of 25 µg/mL. For the analysis, the lyophilized bee venom (3 mg) was dissolved in 10 mL of internal standard. Each sample was filtered through a 0.2 µm PTFE membrane. Bee venom peptide quantification was achieved using calibration curves for apamin (at range 2–60 µg/mL; y = 0.040x + 0.055; R^2^ = 0.999), phospholipase A2 (at range 8–120 µg/mL; y = 0.049x − 0.3562; R^2^ = 0.999), melittin (at range 31–250 µg/mL; y = 0.062x + 0.029; R^2^ = 0.997).

### 2.3. Animals

All experiments in this study were performed on male Swiss mice aged 8 weeks. They were obtained from Janvier Labs Laboratory (ref SN-SWISS). The strain named “RjOrl: SWISS” was created from two albino males and seven outbred albino females with Tyr^C^/Tyr^C^ genotype type. This strain is used in all fields of biomedical research including neuroscience [33]. A total of 80 male Swiss mice with an average body weight of 32.5 g (30–35 g) were used in this study. All animals were maintained on standard laboratory food and tap water ad libitum in a lighting condition of 12 h light/dark cycle. All animal experiments were conducted following the Rules for Animal Care and the Guiding Principles for Experiments Using Animals and were approved by the University of Paris cité Animal Care and Use Committee (agreement number: C750607) “2020”.

### 2.4. Surgical Procedure

During all the surgical procedures, animals were anesthetized by intraperitoneal injection of ketamine 1000 (Vibrac, Carros, France, 100 mg/mL) and xylazine (Rompun 2%). The left buccal and marginal mandibular branches of the facial nerve were exposed via a 2 cm skin incision and sectioned using a micro-scissor (Moria SAV, Antony, France) under a light microscope. Then, the two segments were sutured using an end-to-end epineural suture with 10–0 monofilament (Ethicon; Johnson & Johnson, Issy les Moulineaux, France). The skin was sutured with 4–0 surgical sutures (Ethicon; Johnson & Johnson, Issy les Moulineaux, France) after placing a surgicel^®^ (Johnson & Johnson, Issy les Moulineaux, France), which is used to control the product absorption at the surgical site and to limit the diffusion in adjacent tissues.

### 2.5. Treatments

The animals were split into the groups illustrated in Table 1. In the control group, the facial nerve branches were injured without any treatment. In the BV group, the injured facial nerve branches were treated with local injections of BV solution. In the betamethasone group, the injured facial nerve branches were treated by local injection of betamethasone, and in the bvPLA2 + melittin group, the injured facial branches were treated by local injections of a mixture of bvPLA2 and melittin. Finally, in the sham group, the facial nerve branches were injured, and the normal saline was injected similarly as in the BV-treated group.

Mice received their first injection with BV solution, betamethasone, bvPLA2 + melittin, or normal saline (0.9%) for sham animals immediately after surgery. For administration, BV was dissolved in normal saline (0.9%) and administered by local subcutaneous injection (around the surgical site) for 4 weeks, 3 times a week at the concentration of 0.5 mg/kg for BV, and 2 mg/kg for betamethasone (Celestene 4 mg/mL). Mice treated with a mixture of commercial standard bvPLA2 and melittin (Sigma-Aldrich) by local injection subcutaneous (around the surgical site) at a dose of 0.5 mg/kg (0.25 mg/kg from each standard) for 4 weeks was used to compare the effect of BV and purified bvPLA2 and melittin.

### 2.6. Functional Analysis of Nerve Regeneration

The evaluation of nerve regeneration was performed 2 and 4 weeks after the injury, a time frame that allows for a medium-term analysis of the nerve regeneration process. It consisted first of functional analysis. In rodents, the facial nerve innervates the muscles responsible for the movement of the vibrissae [34]. Thus, the evaluation of the functional recovery was assessed through video recordings of the activity of the vibrissae and nasal deviation.

#### 2.6.1. Whisker Movements

Whisking behavior was video recorded for 2–5 min with a high-speed video camera (Sony HD) during active exploration spontaneously and under stimulation on different devices (Figure 1) as described by Er-Rouassi et al. [35]. A video associated with this article is in the Appendix A.

First device: the mice were placed on a Rota Rod with a bar rotating at four turns per minute. The video film was performed for 2–5 min. The frequency of vibrissae movements of the injured side was compared with the intact side.

Second device: each mouse was placed on a plexiglass bar (39.5 cm long, 17 cm high, and 2 cm wide) with mini platforms (A and B) at the extremities. The animal should walk from point A to point B. This allowed us to compare the whole whiskerpad from arcs A to E and rows 1 to 6. Sometimes, the mice stopped walking in the middle and looked down, with active whisker movements. This allowed us to compare the vibrissae protraction amplitude between both sides.

Third device: the mice were placed in a black tunnel open at the two extremities. Face photography was taken first to evaluate the nasal deviation after surgery. Furthermore, the mice often moved their heads downward under stimulation using a paint brush. In that condition, we can appreciate the movements of maximal protraction of the vibrissae on each side with great precision.

From these recordings, the vibrissae movement was analyzed during the post-surgical period (D1, D3, D7, D10, D14, D16, D25, D30) by two observers blind to treatment. Scores were attributed according to whisker movements on the different devices, on the basis of intermediate scores, as described in Table 2, taking into account scores obtained for each device to obtain a final score [35]. In addition, the protraction/retraction movements were analyzed in more detail using a video capture system (NCH Videopad software, version 12.27).

This scoring system differs from published scores because we considered the separate scores obtained for each device to obtain a final score. The amplitude of whisker protraction was studied in detail, particularly in the third row (C3) “head-down” HD [35].

#### 2.6.2. Measurement of Nasal Deviation

A nasal deviation was immediately observed after surgery. Each mouse was regularly photographed (with a high-speed video camera (Sony HD)) at rest in pre- and post-surgery (at D1, D3, D7, D10, D14, D16, D25, D30, D35, and D38) to quantify the degree of deviation [35]. The tip of the mouse’s nostril and the center of the pupil of both eyes were defined as reference points. By using ImageJ software, version 1.52, the angle α was measured between the midsagittal plane and the line connecting the nostrils as follows:The inter-iris line was drawn linking the center of the right and left pupil.A perpendicular line to the first was then drawn midway between the irises.A line was added connecting the outer edges of the two nostrils.

### 2.7. Retrograde Labeling of Regenerated Facial Motoneurons

The mice were anesthetized as in the initial surgery, and 20 μL of 1% Fluoro-Gold (FG; biovalley, Nanterre, France) was injected into the whiskerpads from both sides at the third row (C3 vibrissa) to detect the regenerated motoneurons in the facial nuclei. Then, 48 h later, the mice were anesthetized, and intracardiac perfusion with 20 mL of normal saline (NaCl 0.1 M, pH 7.2) followed by 50 mL of 4% paraformaldehyde (PFA) was performed. The brainstem was removed and preserved in 4% PFA solution for 24 h. After rinsing in phosphate-buffered saline (0.1 M, pH 7.2), the brainstem was sectioned at a thickness of 30 μm using a vibratome (LEICA VT1000 S, France). The sections were counterstained with NeurotraceTM 530/615 Red Fluorescent Nissl Stain (Thermo Fisher Scientific, Villebon-sur-Yvette, France). They were mounted in Vectashield (Vector Laboratories, Les Ulis, France). Fluorescently labeled motoneurons were identified using an optic fluorescence microscope (NIKON E800). Then, the images were digitalized using the Photometrics CoolSNAPfx CCD camera (Roper Scientific, Lisses, France) and analyzed using the MetaView image analysis software (version 1.52). Afterward, the counting was performed using ImageJ. All retrogradely labeled motoneurons with a visible nucleus were counted through lateral and intermediate facial nuclei on both sides. Only cells with a diameter of at least 20 μm were counted. Counting was performed blindly concerning treatment. Corrections were made for double counting according to Abercrombie’s method using the following formula N = n × {D/(D + d)} (with n being the number of neurons counted; D being the slice thickness; d being the average diameter of a neuron) [36].

### 2.8. Statistical Analysis

All results are presented as the mean ± SD. Comparisons among each group were analyzed by one-way ANOVA followed by *t*-tests using GraphPad Prism Software 8. Values of *p* < 0.05 were considered statistically significant (*** *p* < 0.001; ** *p* < 0.01; * *p* < 0.05).

## 3. Results

### 3.1. UHPLC Peptide Analysis of Honeybee Venom

Using the XSelect CSH130 C18 column, the average retention times for apamin, phospholipase A2, and melittin were 4.63 min, 8.38 min, and 9.68 min, respectively (Figure 2). The maximum concentration was 60 μg/mL for apamin, 120 μg/mL for phospholipase A2, and 250 μg/mL for melittin. The results indicate a linear relationship between the detector responses (peak area) and the concentration of the BV components. The linear correlation coefficient was greater than 0.997 for all compounds studied. The contents of melittin, phospholipase A2, and apamin in our sample of BV were 76.90 ± 0.13%, 11.73 ± 0.13%, and 2.01 ± 0.01%, respectively.

### 3.2. Vibrissae Mouvments

To investigate the potential therapeutic benefit of BV after facial nerve injury, a BV solution at the dose of 0.5 mg/kg (BV group) or betamethasone at 2 mg/kg (betamethasone group) was locally (subcutaneous (s.c)) injected into the injured facial nerve site (three times a week for four weeks). To evaluate the effect of BV on motor recovery of the sectioned sutured facial nerve, we assessed vibrissae movement using video scoring according to the system detailed previously by Er-Rouassi et al. [35]. Immediately after surgery, there was no detectable whisking noted on the injured side, with a caudal retraction of the vibrissae that lasted for 3 days.

Two weeks after injury, analysis of video recordings showed that local BV injections increased the recovery of functional motor performance illustrated by net and asymmetrical vibrissae movements on the injured side (score of 2) group compared to the control group, which still had a score of 1 corresponding to fibrillations (*p* < 0.01) (Figure 3A). However, the effect of betamethasone did not differ from that of control and sham groups. They recovered in the same way as the controls. Furthermore, a slight improvement in motor function was observed in the bvPLA2 + melittin group compared to the controls (*p* < 0.05) (Figure 3A).

Four weeks later, the vibrissae movement recovered a score of 2.5 representing significant voluntary vibrissae motion of the lesioned side relative to the intact side with incomplete recovery of the maximum angle of protraction in the BV group. However, the control group did not recover this score (they still had a score of 2) (*p* < 0.05) (Figure 3B). Statistical analysis revealed no significant differences between the control, betamethasone group, bvPLA2 + melittin group, and sham (*p* > 0.05). These data showed that the local injection of BV improved motor function restoration after section-suture of the buccal and marginal mandibular branches of the facial nerve.

### 3.3. Nasal Deviation

When mice were at rest, the angle α was 89.95 ± 0.02° in intact mice, but it increased to 101.1 ± 0.86° when the buccal and marginal mandibular branches of the FN were injured and were immediately repaired by an epineural suture.

Two weeks after section-suture, the angle α reached 90.24 ± 0.55° in the BV-treated group, which showed a clear effect of BV on the nasal deviation towards the intact side (*p* < 0.001 compared to the other groups). We failed to detect any significant differences between the other groups compared to the control.

Five weeks after the FN section-suture, the angle α of all experimental mice decreased. Statistical analysis revealed a significant difference between control and other groups (BV, betamethasone, and bvPLA2 + melittin groups with *p* < 0.01, *p* < 0.05, and *p* < 0.05, respectively) (Figure 4B).

### 3.4. Fluorescent Retrograde Labeling of Regenerated Motoneurons

Next, we used retrograde labeling to assess the projection patterns of motor axons from the facial nucleus through its buccal and marginal mandibular branches of motor rami. In mice, motoneurons with axons entering the buccal and marginal mandibular branches were localized in the lateral subnuclei with very few labeled cells at the intermediate subnuclei [37] (Figure 5B).

At D14 after injury, a decrease in the labeled cells was observed in all experimental groups (Figure 5C–J). However, in the BV group, more important marked cells were observed (Figure 5D).

Qualitatively, there was an average of 661.16 Fluoro-Gold-labeled facial motoneurons (injected into the whisker pad) in the non-operated group (contralateral side), whereas two weeks after the lesion, a decrease in this average was observed in the control mice at 198.83 (30% of recovery compared to contralateral side). However, an average of 338.33 (53%) was observed in the mice that were treated with BV (*p* < 0.01). A recovery of 40%, which corresponded to an average of 266.6, was observed in the PLA2/melittin group with a significant difference compared to the control, BV, and betamethasone groups (*p* < 0.01, *p* < 0.01, and *p* < 0.001, respectively). After betamethasone treatment at a dose of 2 mg/kg, a mean of 168.5 (27%) was found (*p* < 0.01 and *p* < 0.001 compared to the control and BV-treated group, respectively) (Figure 6A).

At D30, the labeled facial motoneurons were increased in the BV-treated mice group in comparison with the other groups (Figure 5F). A complete recovery was achieved in the BV group, with similar morphology as the intact side (Figure 5F versus Figure 5B). Both betamethasone (Figure 5J) and bvPLA2 + melittin (Figure 5H) groups showed better recovery compared to controls (Figure 5D).

At this stage (D30), the control and BV-treated group had averages count of 254.66 (38%) and 607.33 (93%) labeled motoneurons, respectively (*p* < 0.001). No significant difference was observed between intact and BV-treated mice (*p* > 0.05). The betamethasone-treated group had an average number of 320 (48%) (*p* < 0.05 and *p* < 0.001 compared to the control and BV-treated group, respectively). An average number of 388 (59%) was quantified in the bvPLA2 + melittin group with no significant difference compared to the betamethasone group (*p* > 0.05) (Figure 5B).

## 4. Discussion

This study builds on previous work in which BV and its major components (bvPLA2, melittin, and apamin) demonstrated a neuroprotective effect on neurodegenerative diseases such as Parkinson’s disease, Alzheimer’s disease, and multiple sclerosis, as well as on neurite outgrowth and regeneration after cortical neuron injury [28,38,39].

The results obtained concerning the UHPLC analysis of honeybee venom peptides showed that melittin content was about 76.90% of bee venom peptides. Other studies found an average content of this compound ranged between 60 and 65% and a similar content of apamin and bvPLA A2 [40,41]. It was revealed in a study conducted by Somwongin et al. that the content of melittin in bee venom varied according to *Apis* species, 95.8 ± 3.2% in *Apis dorsata,* 76.5 ± 1.9% in *Apis mellifera,* 66.3 ± 8.6% in *Apis florae,* and 56.8 ± 1.8% in *Apis cerena* [42].

The qualitative and quantitative composition of the peptide fraction of our bee venom analyzed by UHPLC can be used to develop a bee venom standard as an identification method (to establish identity). The results of the quantitative analyses of the individual compounds, namely, melittin, phospholipase A2, and apamin, are intended primarily for use by the pharmaceutical industry.

We investigated here the therapeutic potential of BV therapy in facial nerve injury. The obtained results demonstrated that local injections of BV enhance nerve regeneration in terms of improved functional recovery of the whisker pad.

Indeed, compared to control animals, BV-treated animals had a higher score at D14 and D30. However, vibrissae movements remained inferior compared to the intact side because of incomplete recovery of protraction amplitude. Conversely, the complete disappearance of nasal deviation was seen at D14 in BV-treated mice versus D35 in CTLs. We, therefore, hypothesized that static (nasal deviation) and dynamic (vibrissae movements) deficits followed different time courses.

Along with improved functional restoration, neuronal improvements were observed in mice treated after injection of a retrograde FG marker in the vibrissae. In D30, 38% of the facial motoneurons were connected to the vibrissae muscles. This rate reached 93% after BV treatment. We can therefore deduce that muscle reinnervation by the facial motoneurons is effective and promotes the recovery of vibrissae movements on the injured side. However, it does not lead to optimal recovery of protraction movements on the injured side. Several hypotheses can be suggested to explain the incomplete dynamic functional restoration:

The limiting restoration of facial function may be attributed to the misrouting of axons that regrow and fail to rejoin their original nerve fascicles [11,43]. Each severed axon gives rise to multiple collateral branches, and this excessive branching leads to the reinnervation of different muscles, often with antagonistic functions, for single motoneurons [11].

The neurochemical properties of inhibitory and excitatory inputs are durably altered after facial nerve injury [44,45,46]. These changes may be part of the processes involved in incomplete dynamic recovery. Raslan et al. revealed a positive correlation between functional recovery of the facial nerve (vibrissae protraction amplitude) and glutamatergic and cholinergic deafferentation of facial motoneurons 8 weeks postoperatively [45].

Another factor that requires future attention is the change in the electrophysiological properties of facial motoneurons. After facial nerve injury, a change in the discharge pattern of facial motoneurons has been demonstrated with a decrease in discharge frequency compared to the original one [47,48]. These changes may persist for two months after injury [47]. Thus, these electrophysiological changes in regenerated motor neurons appear to be closely related to the mechanisms controlling functional restoration. Further studies are needed to confirm this hypothesis.

Regarding facial nerve injuries, the most appropriate model to study the effect of BV and potential therapeutic applications in humans seems to be the section-suture of the FN. Indeed, in the crush model of the distal branches of the FN and whatever the duration of the crush (10–30 s), the recovery always occurs at the 9th–10th day, and it is always complete (score 3). The score 2 was observed for only one day. This is not true for the sciatic nerve, where the recovery is observed after 3 weeks [49,50].

A much better recovery than controls was observed in mice treated with a mixture of bvPLA2 and melittin (59% of recovery). Interestingly, bvPLA2 activity can be improved by melittin. A synergistic action between bvPLA2 and melittin has been demonstrated [51]. These two major components of BV showed a minor effect on re-innervation in contrast to the whole bee venom. This can be explained by the involvement of other BV molecules in the regeneration process such as apamin that accelerate neurite outgrowth and axon regeneration after a laceration injury in vitro [28]. It was found that the whole BV and melittin are efficient in the treatment of chemotherapy-induced neuropathy [52]. In addition, Li et al. [53] found that the daily treatment with BV-derived PLA2 for 5 days before the injection of oxaliplatin (a chemotherapy drug that caused neuropathic pain) significantly prevents the development of cold and mechanical allodynia, as well as decreasing the infiltration of macrophages and the pro-inflammatory cytokine (IL-1β) level in the lumbar dorsal root ganglia [53].

The mechanism by which BV and its components act on axonal regrowth has not been established. According to the literature data, several scenarios can be proposed. For instance, it has been demonstrated that PLA2 is expressed in peripheral neurons [54]. Many studies have suggested that PLA2 plays a key role in the process of Wallerian degeneration after PNI, which may be manifested by a rapid trigger of myelin breakdown and the inflammatory cell response [55]. A few hours after a sciatic nerve injury, an expression of PLA2 was observed in the distal segment of the injury that persisted for two weeks. However, no expression of this enzyme was detected after optic nerve injury during the first 3 weeks. This difference correlates directly with Wallerian degeneration in the PNS and CNS. In addition, it has been reported that PLA2 could promote neuronal outgrowth and survival through the production of lysophosphatidylcholine (LPC) [55]. PLA2 and its metabolites, “especially LPC”, could, therefore, be key mediators of the immune cell responses during Wallerian degeneration. Macrophage recruitment leads to rapid clearance of myelin debris, which contains neurite outgrowth inhibitors such as myelin-associated glycoprotein [56]. An absence of macrophage recruitment and severely impaired Wallerian degeneration has been correlated with the lack of expression of PLA2 in the sciatic nerve of C57Wlds mice undergoing Wallerian degeneration [55,57]. It has been shown that LPC injection into the spinal cord of adult mice induces the rapid and transient expression of several immune regulators (e.g., chemokines and cytokines such as MCP-1, MIP-1, GM-CSF, and TNF) [58] that have been shown to mediate macrophage influx and activation, leading to demyelination of dorsal column fibers within four days [57]. The expression of some of these chemokines and cytokines (MCP-1, GM-CSF, and TNF) as well as others such as interleukin (IL)-1 increased in peripheral nerves during Wallerian degeneration [59,60,61]. LPC has also been shown to induce a rapid opening of the blood–brain barrier and increase the expression of the adhesion molecules VCAM1 and ICAM1 in spinal cord endothelial cells, promoting in this way the penetration of inflammatory cells at the site of injury [57]. Both features of LPC are likely to be important for the efficient influx of macrophages into the degenerated nerve.

During PNI, the activation of the immune system creates an environment favorable to nerve regeneration. This discovery could have an impact on future research into nerve regeneration and proposed clinical treatments. The main treatments to date have been powerful anti-inflammatory drugs that may prevent the development of a more favorable environment for regeneration [62]. Betamethasone is one of the synthetic glucocorticoids used pharmacologically as anti-inflammatory and immunosuppressive agents known for their comparable effects to endogenous glucocorticoids. At D30, 48% recovery of facial motoneurons after betamethasone treatment has was, which was statistically significant compared to controls (with 38% of recovery) (*p* < 0.05). Glucocorticoids, therefore, inhibit the effects of TLRs that involve NF-kB. Among the pro-inflammatory cytokines most repressed by glucocorticoid action are IL-lp, 1TL-6, and TNF [63]. The evolution of knowledge about the positive and negative sides of inflammation in the PNI may eventually encourage the development of new therapeutic strategies that may be based more on modulating the intensity of immune system activation, rather than on its repression [62].

Apamin is the third most abundant main element in BV (2% in our sample: see Figure 2). It is probably the origin of the promising effects that we have shown in this study.

A modulation of calcium-dependent potassium channels (SK1, SK2, and SK3) has been described in the axotomized facial motoneurons with an upregulation of SK1 and SK3 mRNAs and a downregulation of SK2 mRNA [64]. SK channels are thought to regulate neuronal excitability by contributing to the slow component of synaptic AHP. However, it has been formally shown that only SK2 is responsible for mAHP using KO mice for each 3 SK subunits [65]. The downregulation of the SK2 mRNA observed in axotomized motoneurons was positively correlated with the diminution of the AHP, which is the cause of an increase in excitability that has been described previously in the axotomized facial motoneurons 8–46 days following the lesion [47]. Neuroprotective effects of apamin on neurons have been recently reported. It may enhance neuronal excitability, synaptic plasticity, and organ potentiation by blocking the Ca^2+^ and activating K^+^ (SK) channels [27,66,67]. Indeed, the application of apamin increases the discharge frequency of tonic neurons.

Recent data claim that apamin is a promising agent of axonal regeneration promoting neurotherapy using primary mature neurons and an in vitro laceration injury model [28]. They could show a greater effect of apamin on axon growth and neurite outgrowth. Interestingly, apamin treatment increased axon growth beyond the borderline at 24 h post-laceration injury. At 48 h, longer neurites were observed with a greater abundance of neuronal axons in the scratched area, which confirmed the long-term positive effect of apamin. In the same model, both BDNF and NGF protein levels were increased by apamin treatment in a dose-dependent manner. Moreover, it has been shown that apamin treatment with an optimal dose increases the F-actin-expressing growth cone, which acts as guidance cues during development on the ends of growing axons. As well as this, an upregulation of neurofilament (NF200), which is one of the regeneration-associated genes, was observed in the apamin group 24 h after laceration. Additionally, the growth-associated protein (GAP43) expression was significantly increased in the apamin group compared with the blank group.

The antioxidant activity can be another way to explain the positive effects of BV. Following PNI, injured motoneurons undergo significant pathological changes in their metabolism, morphology, and electrophysiology [68]. These changes in neural cells promote ROS overproduction. Although few studies have been carried out so far, they have been able to demonstrate the interest of these strategies, as the different antioxidant agents evaluated showed a beneficial effect on nerve regeneration. Following axotomy, a rapid influx of calcium from the injured site occurs. This results in increased calcium concentration at the proximal and distal axon branches, which in turn induces mitochondrial swelling and H_2_O_2_ production [69,70]. H_2_O_2_ then diffuses across the axonal membrane and participates in the activation of neighboring SCs to enable Wallerian degeneration [71].

Lv et al. have found that the antioxidant system was weakened rapidly after acute peripheral nerve injury, and it ceased to regulate Schwann cell plasticity. This weakness was partially dependent on the nuclear factor erythroid 2-like 2 (Nrf2), which causes a temporary state of oxidative stress that is slowly regulated after regeneration [72].

It was revealed that BV components possessed high antioxidant activity [42], and it was found that BV can act on oxidative stress via several mechanisms. The enhancement of endogenous antioxidants such as GSH, SOD, GPx, and CAT is one of these mechanisms [73]. As previously mentioned, in the BV, there are components with antioxidant activity. The efficacy of this activity is usually related to the concentration of melittin, PLA2, and apamin. The antioxidant effects could be caused by the capacity of these compounds to inhibit the lipid peroxidation process and increase superoxidase dismutase activity [74,75], which can be the cause of a rapid elimination of myelin debris, which is an extremely interesting step for regeneration to begin. Moreover, nerve regeneration is dependent on H_2_O_2_, which forms a concentration gradient that attracts leukocytes and then promotes axonal elongation [76]. In addition, melittin regulates the nuclear translocation of Nrf2, leading to increased production of heme oxygenase-1 (HO-1), a major cellular antioxidant enzyme that combats neuronal oxidative stress. Furthermore, melittin also activates the tropomyosin-related kinase receptor B (TrkB)/cAMP-response-element-binding (CREB)/brain-derived neurotrophic factor (BDNF), which allows for a contribution to neuronal neurogenesis and regulation of normal synapse function in the brain [21].

## 5. Conclusions

In conclusion, the present study identified BV as a new therapeutic candidate to treat PNI. The results showed the beneficial effect of an early administration of whole BV in the injured site following a distal facial nerve section-suture. The whisker movement and nasal deviation after BV treatment recovered faster compared to the control and sham groups following a distal section-suture of the facial nerve. In line with functional nerve recovery, normal fluorogold labeling of the facial motoneurons was restored four weeks after surgery, which was never the case in the control group. BV was more efficient than betamethasone and the mixture of bvPLA2 and melittin on both functional and neuronal recovery. However, additional studies in rodents are required to investigate the underlying mechanisms of BV in facial nerve injury and regeneration. We hope that our study will allow the development of new therapies for patients suffering from facial nerve palsy caused by facial injury due to face trauma or pontocerebellar tumor surgery.

## Figures and Tables

**Figure 1 biomolecules-13-00680-f001:**
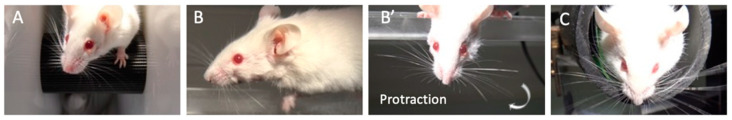
Modality for assessing vibrissae movement. (**A**) Video on Rota Rod; (**B**) and (**B’**) video on a plexiglass bar; (**C**) video using a tunnel.

**Figure 2 biomolecules-13-00680-f002:**
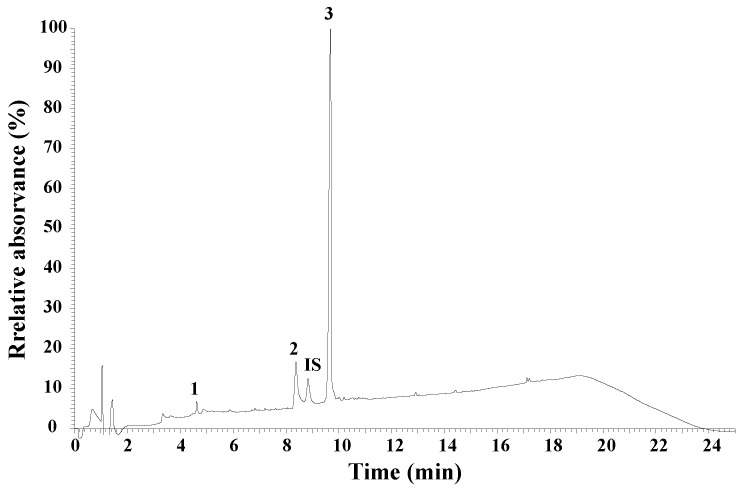
Chromatographic profile of Morocco bee venom: 1—apamin; 2—phospholipase A2 (PLA2); 3—melittin; IS—internal standard (cytochrome c, 25 µg/mL).

**Figure 3 biomolecules-13-00680-f003:**
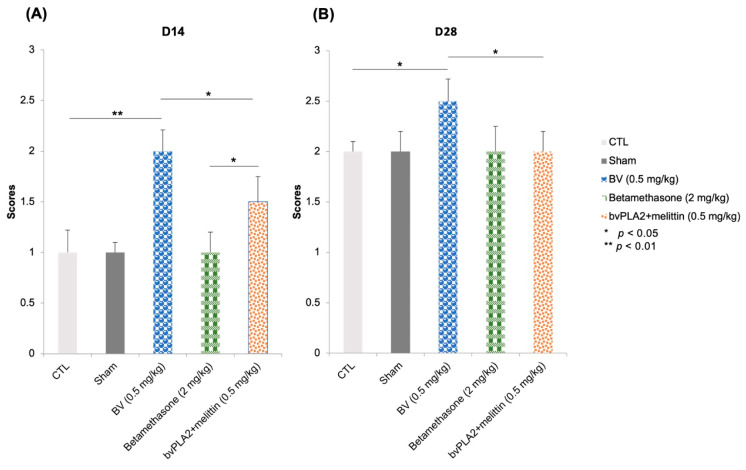
Graphical view of the mean and the standard deviation of the functional analysis (whisker movement). (**A**) Whisker movement 14 days after distal section-suture of the buccal and marginal mandibular branches of the facial nerve in different groups: control (injured mice without treatment), sham (saline 0.9%), BV (0.5 mg/kg of bee venom), betamethasone (2 mg/kg of Celestene), bvPLA2 + melittin (0.5 mg/kg of a mixture of bvPLA2 and melittin). (**B**) Whisker movement 28 days after distal section-suture of the buccal and marginal mandibular branches of the facial nerve in the same groups: control, sham (0.5 mg/kg of saline), BV (0.5 mg/kg of bee venom), betamethasone (2 mg/kg of Celestene), bvPLA2 + melittin (0.5 mg/kg of a mixture of bvPLA2 and melittin). The data were tested by one-way ANOVA followed by *t*-tests.

**Figure 4 biomolecules-13-00680-f004:**
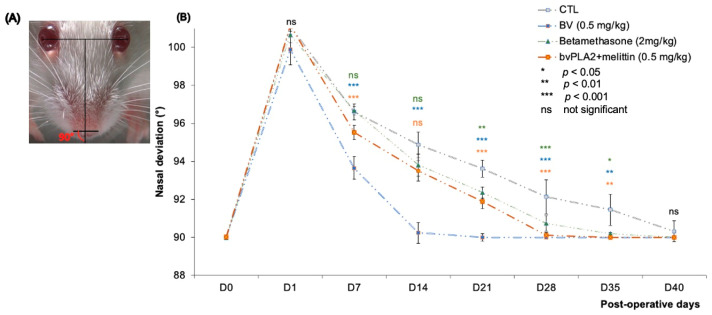
(**A**) The angle α was measured between the midsagittal plane and the line connecting the nostrils. (**B**) Measurement of the angle α at different postoperative days after FN branches SS. The difference among the groups was analyzed by *t*-test. Compared with the BV-treated groups, the control group had significantly lower values with the measurement (*** *p* < 0.001, ** *p* < 0.01). Note that a return to the normal nasal position was observed on D14 in the BV treated group (*** *p* < 0.001) versus D40 in the control group.

**Figure 5 biomolecules-13-00680-f005:**
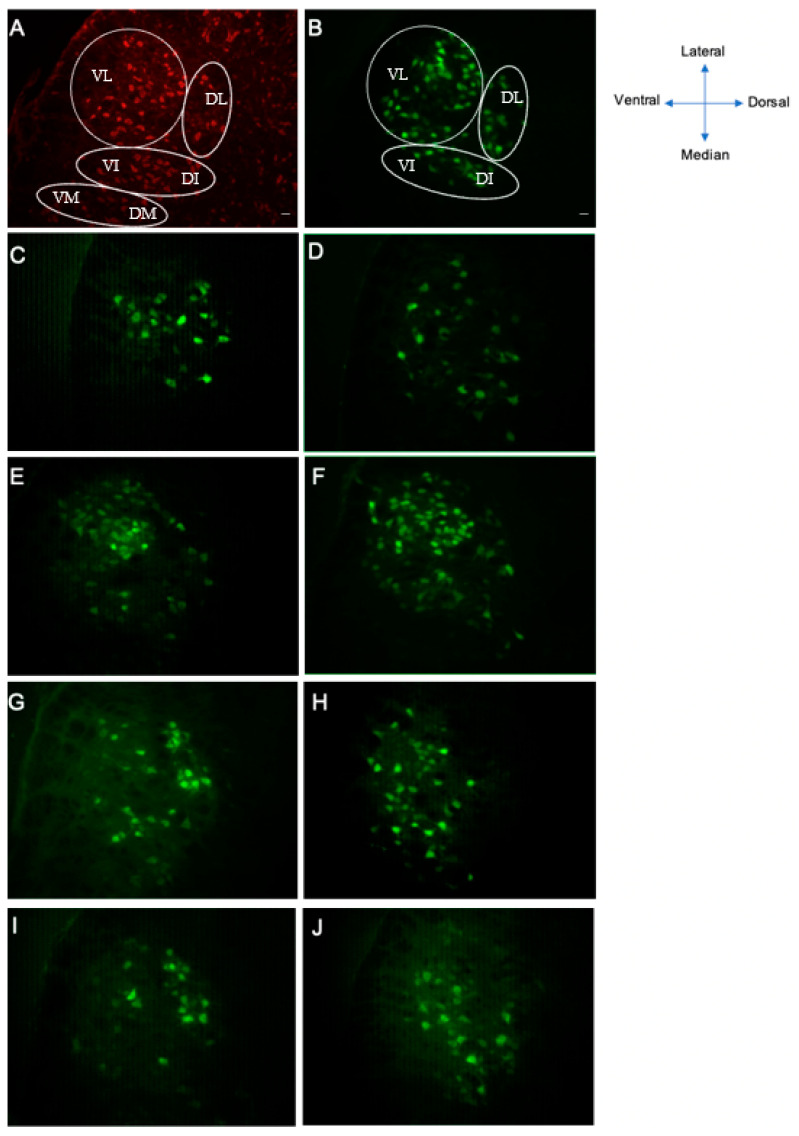
The precision of reinnervation 2 and 4 weeks after SS of the buccal and marginal mandibular branches of the facial nerve. (**A**) Nissl-like fluorescence counterstain “Neurotrace”; (**B**) FG retrograde labeling of facial motoneurons of the contralateral side (noninjured). (**C**,**D**) Facial nucleus after injection of FG into the whisker pads after surgery in control group; (**C**) 2 weeks after surgery and (**D**) 4 weeks after surgery. (**E**,**F**) Facial nucleus after injection of FG into the whisker pads in the BV -treated group; (**E**) 2 weeks after surgery and (**F**) 4 weeks after surgery. (**G**,**H**) Facial nucleus after injection of FG into the whisker pads in the bvPLA2 + melittin group; (**G**) 2 weeks after surgery and (**H**) 4 weeks after surgery. (**I**,**J**) Facial nucleus after injection of FG into the whisker pads in the betamethasone group; (**I**) 2 weeks after surgery and (**J**) 4 weeks after surgery. Retrograde labeling with Fluoro-Gold was confined to the lateral subnuclei with very few labeled cells at the intermediate subnuclei. VL, ventrolateral; DL, dorsolateral; VI, ventral intermediate; DI, dorsal intermediate; VM, ventromedial; DM, dorsomedial. Scale bar = 20 μm.

**Figure 6 biomolecules-13-00680-f006:**
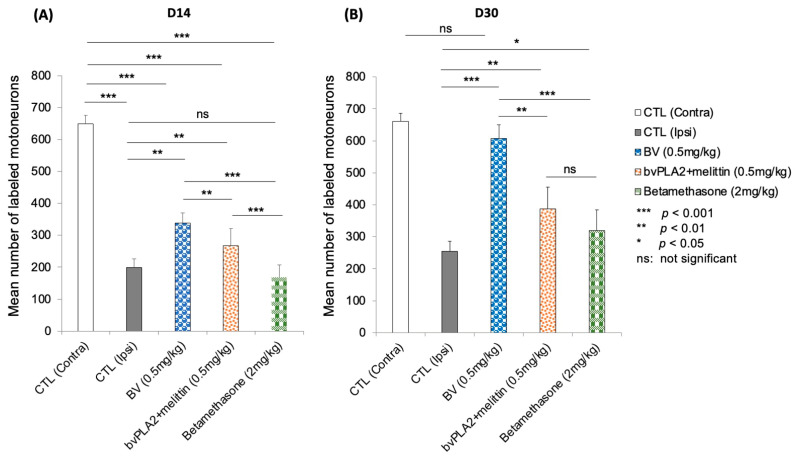
Quantitative analysis: Retrograde-labeled neurons were quantified 2 weeks after facial nerve injury. The data are presented as the means ± SD, and differences were tested by *t*-test. (**A**) On D14, the mean number of laterals labeled facial motoneurons was decreased, consistent with facial nerve section-suture. The number of neurons in the BV group was higher than that in the control and betamethasone groups (*** *p* < 0.001 and ** *p* < 0.01, respectively). (**B**) On D30, the number of labeled motoneurons returned to normal level in the BV-treated group (no statistical difference was observed compared to the contralateral side *p* > 0.05), which was not the case for the other experimental groups.

**Table 1 biomolecules-13-00680-t001:** Experimental design of the study.

Groups	Treatment	Behavioral Study	Retrograde Fluorogold Labeling	
		D1 to D30	D14	D30
Control	Injury without treatment	*n* = 6	*n* = 6	*n* = 6
Sham group (saline)	Injury + normal saline injection at the injury site (0.5 mg/kg)	*n* = 6	/	/
BV	Injury + BV injection at the injury site (0.5 mg/kg)	*n* = 6	*n* = 6	*n* = 6
bvPLA2 + melittin	Injury + bvPLA2 + melittin injection at the injury site (0.5 mg/kg)	*n* = 6	*n* = 6	*n* = 6
Betamethasone	Injury + betamethasone injection at the injury site (2 mg/kg)	*n* = 6	*n* = 6	*n* = 6

The control group, BV group, betamethasone group, bvPLA2 + melittin group, and sham group (saline 0.9%) were studied after FN section-suture. For the behavioral study, the number of mice used is indicated for each group. Similarly, in the morphological, study, the number of mice is indicated for D14 and D30.

**Table 2 biomolecules-13-00680-t002:** Videoscoring of whisker movement recovery.

Rota Rod Score	Bar Score	Head-Down Score	Tunnel Score	Final Score
0	0	0	0	0: No movement
1	1	1	1	1: Slight vibrissae movements
2	2	1	1	1.5: Slight fibrillations on the injured side compared to the intact side on some devices
2	2	2	2	2: Net and asymmetrical voluntary movement of the lesioned side (smaller amplitude) relative to the intact side on all devices
3	3	2	3	2.5: Net and asymmetrical voluntary movement of the lesioned side relative to the intact side for some, but not on all devices (incomplete amplitude)
3	3	3	3	3: Normal and symmetrical movement in the injured side compared to the intact side (complete amplitude and frequency)

## Data Availability

Data are available upon request.

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
