# Peer review of "Beneficial Effect of Bee Venom and Its Major Components on Facial Nerve Injury Induced in Mice"

_biomolecules, 2023, doi:10.3390/biom13040680_

Round 1

Reviewer 1 Report (New Reviewer)

This is a well written manuscript concerning the potential of bee venom in the regeneration of facial nerve injury. The study was solely based on the neurobehavior and retrograde motoneuron labelling. The supporting data seems a little bit weakness. I wondered why the authors did not take the facial nerve when in the animal sacrifice for determination of axon count or axon thickness as well immunohistochemistry in S-100, neurofilament, or CD 68, which easily approved the hypothesis of antioxidant effect of bee venom on the recovery of facial nerve suggested by the authors.

In addition, in Figure 5, the illustration should follow the rules of alphabetic sequence from A to J. There was missing in figure C depiction.

Author Response

We  thank the editor and reviewers for helping to improve our paper. The responses were included in the manuscript according to the suggestions of the reviewers, and every revision made is highlighted in blue.

Reviewer 2 Report (New Reviewer)

I read the paper entitled " Beneficial Effect of Bee Venom and its Major Components on 2
Facial nerve Injury Induced in Mice" by Er-Rouassa H et al. The paper describes the positive therapeutical effects of Bee Venom (BV) administration by injection in the treatment of peripheral nerve injuries and the recovery of normal nerve function.

The paper is interesting showing the potency of new drugs derived from natural compounds and it could be of interest to a wide variety of readers, from pharmacologists to neurologists as well as basic scientists.

I have some concerns about the study. Usually, peripheral nerve injuries are following a wide spectrum of lesions, from crushing lesions with no axotomy to cutting lesions with axotomy to nerve strain lesions. These aspects, together with the ability of surgeons to connect the broken nerve stumps are a major point for nerve function recovery. Moreover, we have peripheral lesions which can be easily reached (for instance hand or legs) and others which are not easily reached (for instance the auto compression of the facial nerve in the facial bone canal during a facial nerve inflammation). Finally, there are acute peripheral nerve injuries and chronic nerve injuries, some of them are a combination of an inflammatory reaction with nerve compression in a particular anatomical site (see facial nerve above).

Therefore I wonder if the results of the utilized experimental approach could be useful in all these different injuries both from the physiopathological point of view and the temporal point of view. Or, on the contrary, these results could be better focussed on a specific type of PNI like the lesions in which the nerve is cut and nerve stumps are re-connected by the surgeon, in a view in which the control of inflammation and stimulation of physiological nerve repair mechanisms are a necessary side point for a good surgical result.

I think that these aspects should be carefully and frankly discussed.

Therefore I consider the paper suitable for publication in this Journal after major revisions.

Author Response

We  thank the editor and reviewers for helping to improve our paper. The responses were included in the manuscript according to the suggestions of the reviewers, and every revision made is highlighted in blue.

Reviewer 3 Report (New Reviewer)

Dear Authors,

The submitted article is very interesting and valuable. The described study shows the potential effect of bee venom on functional recovery and reinnervation after facial nerve section-suture in an animal model. Take into consideration, that peripheral nerve injury constitutes a health problem among patients all over the world, so this pre-clinal study seems important.

The chapter Introduction is well developed. However, In introduction in line 76 Authors should add references. I think they missed it.  Authors presented three aims of conducted study.

In chapter – Material and Methods, in line 100 Authors should correct the numeration of subchapter on 2.2.1. and enter a capital letter.

All subchapter 2.2. should have change the numeration. In lines 128 and 129 Authors should enter “ at range” before range.

The chapter - Results is well described. I suggest to correct Figure 4. The mentioned Figure has low resolution.

The chapter - Discussion is correctly carried out with reference to numerous literature data.

The chapter-  Conclusion should be refreshed and better relate to the purpose of the research. Moreover, in conclusion the reader should find the answer to three major aims presented in introduction.

Author Response

We  thank the editor and reviewers for helping to improve our paper. The responses were included in the manuscript according to the suggestions of the reviewers, and every revision made is highlighted in blue

Round 2

Reviewer 2 Report (New Reviewer)

I read the Authors replies to my comments and I think that now the paper can be accepted for publication

This manuscript is a resubmission of an earlier submission. The following is a list of the peer review reports and author responses from that submission.

Round 1

Reviewer 1 Report

Hafsa Er-Rouassi et al., conduct a study to see the Beneficial Effect of Bee Venom and its Major Components on Facial nerve Injury Induced in Mice. The objective of the study is interesting. However, there are many flaws in the experimental design or results with molecular results. The following are the main concerns that the authors can be addressed.

 1.      It is not clear how the treatment is injected locally. Please clarify how it was injected into the nerve injury site three times/per week. Was it injected around the nerve injury? The volume of injected dosages is also not clear.  What was the logic to select these doses, if it is a novel approach? Did the authors conduct a pilot study on therapeutic dosage selection? If so, please provide it in supplementary. Why did authors select Betamethasone to compare BV or bvPLA2+melittin, though it is not a therapeutic choice for nerve injury?

2.      Please provide details of the physiological serum (source/species/strain). There is a missing un-injured control group (abstract vs manuscript text).  

3.      The therapeutic effect of different groups was compared with the functional recovery of animals. Though the data is promising, there is a missing molecular analysis. What was the BV molecular signaling to heal nerve transection and suture injury?

4.      Authors need to provide nerve histomorphometric (eg: H and E, Masson's trichrome (fibrosis) staining) and targeted biochemical-molecular analysis (eg: western blotting or qRTPCR gene expression) for example neurogenesis, anti-inflammatory, anti-oxidative stress, etc after addressing the above concerns. It is a missing specific aim.

5.      The retrograde analysis of motor nerve recovery is a good technique.

6.      In Figure 2: The difference between the control and sham groups is not clear. Is CTL an uninjured group?  If so, why the score is low as compared to sham (injured +no treatment) or other groups? Why the score of BV is significantly upregulated as compared to CTL (uninjured nerve)? What is the score of uninjured nerves?

7.      Please mention the body weight of the animals.  

8.      In a nutshell, it is an interesting approach, however, the depth of the investigation is not convincing to understand the therapeutic benefits of BV. Thus, a further extended investigation is required to consider for publication. 

Reviewer 2 Report

The authors studied the “Beneficial Effect of Bee Venom and its Major Components on Facial nerve Injury Induced in Mice”. The animal model in mice is the distal section-suture of facial nerve branches following therapies with Bee Venom and various combinations of BV components. The results indicate that BV injections enhanced functional and neuronal outcomes after PNI.

In principle the topic of the study is interesting.

Unfortunately, the manuscript is poorly written and a lot of uncertainties are evident. Especially, the experimental design and the exact performance of the experiments are unclear. When did the injections of the components start after surgery, were exactly and how were the injections near the wound placed (injury site), the author described it as local or subcutaneous or into the neuronal suture site, where the mice anesthetised to do so? The doses of the BV were 0.5 mg/kg bw and the same was used for bvPLA2+Melittin (do you mean each that of the components was in that dose? – then it would be much more than the BV) - how do you find these doses, why was betamethasone used and why in the used dose?

Why do you tell the reader x-fold in the manuscript the doses used in the various groups – it would be sufficient to say it once in the M+M, and then use a strict naming of the animal groups throughout the manuscript.

It is not scientifically correct to used a table (here table 3), that without changing a single word can be found as table 2 in “Er-Rouassi, H.; Benichou, L.; Lyoussi, B.; Vidal, C. Efficacy of LED Photobiomodulation for Functional and Axonal Regeneration After Facial Nerve Section-Suture. Front. Neurol. 2022, 13, 827218, doi:10.3389/fneur.2022.827218.” In the present manuscript the data are not understandable and completely useless.

Further questions

Line 19: what is a distal section – please provide a picture or a schematic drawing to show the reader what mean

Line 24: what is “physiologic serum” 

Line 69: what is sweet BV

Line 75: this sentence gives no sense

Line 87: what is the difference between control mice and sham mice – which differences would you expext besides the stress factor of the 3x4 injections in the shams

Line 92: how was the BV collected?

Line 104: please provide the date of permission

Line 132: what is the role of the “soaked surgicel with the corresponding product (according to the corresponding group) placed at the surgical site”

Line 162: “Whisking behavior was video recorded for 2 -5 min during active exploration on dif-ferent devices as described by Er-Rouassi et al. [27].” My comment: The reader should not be forced to read your previous paper to catch the point.

Line 214: the table is useless; it is completely said in the text just above the table 3

Line 223: I hear the first timer of the 48 h activity – what was performed at that time point

Line 231: what is a “sharp and asymmetrical vibrissae movement”

Line 240: ”this score (they still had a score of 2)” – is 2 right in this sentence? Should it be 1?

Line 248: in the figure A and B are missing, and why is the order of the various groups another than in table 1 – this is really confusing

Line 267: to which group comparisons do the p-values belong

Line 271: the picture (A) is exactly found in “Whisking behavior was video recorded for 2 -5 min during active exploration on different devices as described by Er-Rouassi et al. [27].” Even with the exact angle

Line 286 – Figure 4: please provide the encircling of the subnuclei in all pictures C-J and name the subnuclei of these nice experiments

Line 319, figure 5: what do you mean .. the mean number of laterals labeled facial motoneurons.

Reviewer 3 Report

The authors evaluate the potential effect of bee venom (BV) and its major components in a model of Peripheral nerve injury (PNI) in the mouse. For that, the BV used in this study was analysed using UHPLC.

This manuscript is interesting and the study is new; nevertheless needs substantial improvements and corrections before publishing may be possible.

General points:

Please add a list of abbreviations before References section to your manuscript.

Special points:

Keywords: please add also to keywords: Swiss mice.

Please add to your manuscript a time-line of all your experiments as a Figure 1.

 Introduction

Lines 39-50: Please add multiple references at the end of each these sentences.

Lines 45-50: Please describe exactly all these studies and treatment possibilities.

Lines 54-55: Please add multiple references at the end of this sentence.

Lines 65-68: Please add multiple references at the end of this sentence.

Lines 71-72: Please add multiple references at the end of this sentence.

Lines 72-81: Please describe exactly all these studies.

Lines 82-84: Please describe this more exactly and please add an appropriate references.

Materials and Methods

Please add the exactly number of the animals used for each method.

Please add the appropriate references for each section.

Line 99: please add the gender and the exactly total number of the animals used in all your experiment.

Please add also the exactly description of the Swiss mice line with appropriate literature.

Lines 102-104: please add also the number and the date of the permission for all your experiments.

Lines 147-148: please add an exactly time point of the first injection with BV solution.

2.6. Functional analysis of nerve regeneration

Please describe this method more exactly and please add a video for each experimental group to the Supplement.

2.6.1. Whisker movements

Please describe this method more exactly and please add a video for each experimental group to the Supplement. Please describe the study number 27 more exactly.

Table 2: please describe all these scores exactly in your manuscript text and add the appropriate references.

2.6.2. Measurement of nasal deviation

Please describe this mezhod more exactly. Which camera exactly you used for that? Please add.   

2.7. Retrograde Labeling of Regenerated facial Motoneurons

Please describe more exactly the counting.

Results

Lines 220-226: please describe more exactly the reference number 27.

Discussion

Lines 321-324: please describe more exactly all these studies.

Lines 341-342: please describe more exactly the reference number 27.

Lines 387-388: Please add multiple references at the end of this sentence.